# Wheat-*Psathyrostachys huashanica* 4Ns Additional Line Confers Resistance to Fusarium Head Blight

**DOI:** 10.3390/plants14071104

**Published:** 2025-04-02

**Authors:** Yinghui Li, Hang Peng, Hao Zhang, Liangxi Li, Muhammad Saqlain, Dandan Wu, Wei Zhu, Lili Xu, Yiran Cheng, Yi Wang, Jian Zeng, Lina Sha, Haiqin Zhang, Xing Fan, Yonghong Zhou, Houyang Kang

**Affiliations:** 1State Key Laboratory of Crop Gene Exploration and Utilization in Southwest China, Sichuan Agricultural University, Chengdu 611130, China; liyinghui@sicau.edu.cn (Y.L.); 13808181840@163.com (H.P.); m18200268660@163.com (H.Z.); 15390084469@163.com (L.L.); 2024512001@stu.sicau.edu.cn (M.S.); wudandan@sicau.edu.cn (D.W.); zhuwei202209@163.com (W.Z.); chengyiran@sicau.edu.cn (Y.C.) wangyi@sicau.edu.cn (Y.W.); fanxing9988@163.com (X.F.); zhouyh@sicau.edu.cn (Y.Z.); 2Triticeae Research Institute, Sichuan Agricultural University, Chengdu 611130, China; xulili_0627@126.com; 3College of Resources, Sichuan Agricultural University, Chengdu 611130, China; zengjian@sicau.edu.cn; 4College of Grassland Science and Technology, Sichuan Agricultural University, Chengdu 611130, China; rice_shazhi@163.com (L.S.); haiqinzhang@163.com (H.Z.)

**Keywords:** Fusarium head blight, *Psathyrostachys huashanica*, 4Ns additional line, wheat resistance breeding

## Abstract

Fusarium head blight (FHB) is one of the major wheat diseases caused by Fusarium species (mainly *Fusarium graminearum* and *Fusarium asiaticum*), resulting in significant global wheat yield losses and risks to food security. Breeding wheat varieties with resistance genes is the most environmentally friendly and economical strategy for controlling FHB. *Psathyrostachys huashanica* Keng ex P. C. Kuo (2*n* = 2*x* = 14, NsNs), which showed abiotic tolerance and biotic resistance, has significant research value and potential as an important genetic resource for wheat improvement. In previous studies, we crossed *Psathyrostachys huashanica* with common wheat and developed wheat lines containing different N_S_ chromosomes. In this study, we identified a 4N_S_ additional line, DA26, from the progenies of wheat-*P. huashanica*-derived lines using genomic in situ hybridization (GISH) and fluorescence in situ hybridization (FISH) analyses. Line DA26 showed high resistance to Fusarium head blight (FHB) in the greenhouse and field conditions. However, the parental common wheat lines Chinese Spring (CS) and CS*ph2b* mutant showed high susceptibility to FHB. A field evaluation of the agronomic traits showed that the plant height of DA26 was significantly lower than CS, while there were no significant differences in the other agronomic traits. In addition, we also developed eight 4Ns-specific primers to identify the 4Ns chromosome, which can facilitate wheat breeding and FHB resistance gene mapping in the future.

## 1. Introduction

Wheat is an important crop as a main food source worldwide, providing humans with energy, essential nutrients, vitamins, and minerals [1]. Fusarium head blight (FHB) is one of the most destructive diseases of wheat worldwide. It is caused by Fusarium species, mainly *Fusarium graminearum* and *Fusarium asiaticum*, which primarily infect developing wheat spikes and affect the development of the spikes and grains. FHB epidemics have spread to many wheat-growing areas worldwide and have been considered a major threat to wheat production since the early 20th century [2]. FHB epidemics are causing high yield losses, reaching 70% in some epidemic years, and reducing grain quality severely due to toxins such as deoxynivalenol (DON) and Moniliformin (MON) [3,4,5]. This grain contamination with mycotoxins severely threatens human and livestock health, causing headaches, abdominal pain, dizziness, fever, and sleepiness [6]. Developing and utilizing resistant cultivars are the most economical and environmentally friendly strategy to control this disease.

The resistance of FHB in plants is very complex and controlled by quantitative trait loci (QTL), which might be affected by environments. Therefore, it is challenging to identify and clone stable FHB resistance QTLs. Until now, nine FHB resistance QTLs (*Fhb1*-*Fhb9*) have been officially documented from common wheat or wild wheat relatives [7]. So far, only *Fhb1* from Chinese wheat cultivar Sumai 3 and *Fhb7* from *Thinopyrum ponticum* have been cloned, which encode a histidine-rich calcium-binding protein and glutathione S-transferase, respectively [8,9,10]. These two genes have been used in wheat breeding systems and deployed in several wheat varieties, such as Ningmai 13, MS INTA 416, Shannong 48, and Zhongke 166, etc. [11,12,13,14]. In addition, more than 500 QTLs have been identified from different populations or environments, controlling the FHB resistance at different levels. However, most identified QTLs were minor disease resistance genes and not stable in different genetic backgrounds or environments, and only a few of them have been confirmed that could be used in wheat breeding [7]. Therefore, identifying novel genetic resources for high resistance to FHB is an ongoing task for wheat resistance breeding.

Wild wheat relatives are important genetic resources, and many successful introgressions from wild wheat species have been deployed into common wheat varieties, contributing to wheat disease resistance and yield production in the field, such as wheat–rye 1RS·1BL translocations and wheat–*Dasypyrum villosum* translocation T6VS·6AL [15,16]. *Fhb3*, *Fhb6*, and *Fhb7* have been identified from *Leymus racemosus*, *Elymus tsukushiensis*, and *Th. ponticum*, respectively, highlighting the important role of these wild wheat relatives on wheat FHB resistance improvements. Moreover, *QFhb.Er-7StL* identified from *Elymus repens* and *FhbRc2* located on *3ScL* of *Roegneria ciliaris* have been introduced to the common wheat background through the introgression lines contributing to the resistance to FHB [17,18]. Chromosomes 7E of diploid and tetraploid *Th. elongatum* is also used to improve the resistance to FHB in common wheat [11,19]. These studies support the importance of tertiary genetic pools for wheat FHB resistance.

*Psathyrostachys huashanica* Keng ex P. C. Kuo (2*n* = 2*x* = 14, NsNs) is a perennial species of the genus *Psathyrostachys* Nevski. It is restricted to the Huashan Mountain region of Shaanxi province, China. It has significant, valuable potential for wheat breeding and many useful agronomic traits, such as early maturation, multiple kernels and tillers, abiotic tolerance, and biotic resistance [20,21,22,23,24,25]. In previous studies, we crossed the Chinese Spring *ph2b* (CS*ph2b*) mutant with *P. huashanica*, which facilitated homoeologous pairing between alien chromosomes and wheat chromosomes and improved the fertility rate of F_1_ hybrids [26]. We developed a series of derived lines involving different Ns chromosomes by backcrossing the wheat–*P. huashanica* F_1_ hybrids with common wheat Chinese Spring (CS). In this study, we selected a 4Ns additional line DA26 from the F_6_ generation of CS*ph2b*/*P. huashanica*//CS. The main objectives of our study are as follows: (1) characterize the chromosomal composition of DA26 via in situ hybridization, (2) evaluate the FHB resistance and agronomic traits of DA26, and (3) develop molecular markers specific for the alien 4Ns chromosome of DA26, which can efficiently trace the 4Ns chromosome or chromosomal segments of *P. huashanica* in the wheat background. The newly developed germplasm resource will be potentially useful for wheat resistance breeding.

## 2. Materials and Methods

### 2.1. Plant Materials

*Psathyrostachys huashanica* Keng (2*n* = 2*x* = 14, NsNs) accession ZY3156 was collected from the Qinling Mountains in Shaanxi Province, China, by Profs. C. Yen and J.L. Yang of Sichuan Agricultural University. Chinese Spring (CS, 2*n* = 6*x* = 42, AABBDD) and CS*ph2b* mutant were common wheat (*Triticum aestivum* L.) lines. The wheat–*P. huashanica* 4Ns disomic addition line DA26 was derived from the F_6_ generation of CS*ph2b*/*P. huashanica*//CS. Canadian wheat cultivar Roblin (2*n* = 6*x* = 42, AABBDD) was used as a susceptible control for the assessment of FHB resistance, while a Chinese wheat cultivar Sumai 3 (2*n* = 6*x* = 42, AABBDD) was used as a resistant control. The related species of wheat, *Leymus racemosus* (accession ZY 07023, 2*n* = 4*x* = 28, NsNsXmXm), *Campeiostachys kamoji* (syn. *Elymus tsukushiensis*, accession ZY 230367, 2*n* = 6*x* = 42, StStHHYY), and *Thinopyrum ponticum* (accession PI 531737, 2*n* = 10*x* = 42, EeEeEbEbExExStStStSt), were used as positive controls for the FHB resistance genes *Fhb3*, *Fhb6*, and *Fhb7*, respectively. All these materials are stored in Triticeae Research Institute, Sichuan Agricultural University.

### 2.2. Cytogenetic Analysis

Wheat root tips were incubated with nitrous oxide gas at 22 °C for 150 min, followed by fixation in glacial acetic acid for 5–10 min. The meristems were cut and digested with pectinase and cellulase (Yakult Pharmaceutical Ind. Co., Tokyo, Japan). The slide preparation for GISH and mc-FISH analyses was described previously [27]. The genomic DNA of *P. huashanica* was labeled via the nick translation method with an Atto550 NT labeling kit (Jena Bioscience, Jena, Germany), serving as the GISH probe. Genomic DNA of CS was used as a blocker; the probe-to-blocker ratio was 1:150. Chromosomes were counterstained with 4,6-diamino-2-phenylindole solution (DAPI; Vector Laboratories, Burlingame, CA, USA), and fluorescence signals were detected and visualized using a fluorescence microscope (Olympus BX63) equipped with a Photometric SenSys DP-70 CCD camera (Olympus, Tokyo, Japan).

After the GISH analysis, the photographed slides were washed successively with 75% and 100% (*v*/*v*) ethanol for 5 min each, then with 2× SSC (Sulfide Stress Corrosion Cracking) at 60 °C for 30 min, and again with 75% and 100% (*v*/*v*) ethanol for 5 min each and then exposed to bright light for 48 h to eliminate GISH signals. The genomic DNA of *P. huashanica* was used as the GISH probe DNA. In the FISH experiments, we used oligonucleotide probes pSc119.2 (labeled with 6-FAM at the 5′ end) and pTa535 (labeled with TAMRA at the 5′ end), synthesized by Sangon Biotech (Chengdu, China) Co., Ltd. to distinguish individual wheat chromosomes in DA26. The synthesis of these probes is described in Tang et al. (2014) [28]. To determine the homologous group of the alien chromosomes, we referred to the method described in Zhang et al. (2022) [27] and used three probes, Oligo-44, pTa71A-2, and pSc200, to identify the alien chromosomes in DA26. The sequence of each probe is listed in Appendix A.

### 2.3. Evaluation and Statistical Analysis of FHB Resistance

The FHB responses of wheat lines were evaluated at a controlled greenhouse and the field at Wenjiang Experimental Station of Sichuan Agricultural University using a single-spikelet inoculation method. Wheat plants were grown under a daily cycle of 16 h of light at 23 °C and 8 h of darkness at 20 °C with a relative humidity of 75% in a greenhouse infected with *Fg* strain PH-1 (NRRL 31084). In the field, we infected the wheat plants with *Fg* strain PH-1 and mixed Sichuan *Fg* isolates [29]. The percentage of infected spikes (PISs) was recorded and photographed at 21 days post-inoculation (dpi). According to the Chinese technical specification for FHB resistance evaluation (NY/T 2954-2016 [30]), FHB severity was classified into four levels (1~4) based on PIS and rachis diseased symptoms, with resistance level further graded as R (Resistant), MR (Moderately Resistant), MS (Moderately Susceptible), and S (Susceptible) based on the calculated severity.

### 2.4. Detection of Known FHB Resistance Genes from Wild Relatives

The markers of three known *Fhb* genes from wild relatives of wheat, including *Fhb3*-linked marker BE585744 developed by Qi et al. (2008) [31], *Fhb6* functional marker BE598612 developed by Cainong et al. (2015) [32], and the *Fhb7* functional marker *GST* developed by Wang et al. (2020) [8], were employed to detect whether DA26 carries *Fhb3*, *Fhb6*, and *Fhb7* genes, respectively. The PCR amplification mixture had a final volume of 20 μL, which consisted of 1 μL of template DNA with a concentration of 200 ng/μL, 10 μL of 2× Taq Master Mix for PAGE (Dye Plus, Thermo Fisher Scientific, Rastatt, Germany), 1.0 μL of each primer with a concentration of 10 μM, and 7.0 μL of ddH_2_O. The PCR program was as follows: initial denaturation at 94 °C for 5 min, followed by 35 cycles of denaturation at 94 °C for 30 s, annealing at 58 °C for 30 s, and extension at 72 °C for 30 s, with a final extension at 72 °C for 10 min. For *Fhb6*, the PCR products were digested with *HaeIII* (Thermo Fisher Scientific, Rastatt, Germany) for 30 min after amplification. The enzyme digestion reaction system and procedure followed the manufacturer’s instructions. PCR products were detected using 3% agarose gel electrophoresis.

### 2.5. Agronomic Trait Evaluation

During the 2023–2024 growing season, the morphological traits of the line DA26 and its wheat parents were assessed in a field experiment with three repetitions at the Wenjiang Experimental Station of Sichuan Agricultural University in Chengdu, China. For each repetition, 15 seeds of each line were sewn in rows 1.5 m long and spaced 0.3 m apart. At the physiological maturity stage, 15 randomly selected plants of each line were harvested to evaluate their morphological traits, comprising plant height, tiller number, spike length, number of spikelets per spike, number of kernels per spike, and thousand-grain weight. The IBM SPSS Statistics 24.0 software suite analyzed the remarkable differences in all measured traits between DA26 and its wheat parental lines.

### 2.6. Development and Verification of PCR-Based Markers

Using the published whole-genome sequence of *Leymus chinensis* (Lc6-5) [33], we selected sequences from the 4Ns chromosome and filtered forward EVM gene sequences based on the annotation file. Through the TriticeaeGeneTribe, a homology database for the Triticeae tribe (wheat, durum wheat, barley, and their relatives) (www.cau.edu.cn), we performed collinearity analysis and selected genes with strong collinearity and specificity to the *L. chinensis* genome. Primers targeting 100–300 bp fragments were designed using Primer 3 Plus (Primer3Plus—Pick Primers). The designed primers were validated through in silico PCR on the Wheat Multi-Omics website (http://wheatomics.sdau.edu.cn/). All validated primers were synthesized by Sangon Biotech (Shanghai, China). The amplified products were electrophoresed in 3% agarose gel. The PCR amplification system was the same for detecting disease resistance genes described above. The PCR protocol was as follows: 94 °C for 5 min, followed by 34 cycles of 94 °C for 30 s, 60 °C for 30 s, and 72 °C for 30 s, and a final extension at 72 °C for 10 min. The sequence of each marker is listed in Appendix A.

## 3. Results

### 3.1. Identification of Wheat–Psathyrostachys huashanica 4Ns Additional Line

To check the chromosomal composition of these progenies, we used GISH analysis and identified a monosomic addition line containing 43 chromosomes, one of which might be a chromosome from *P. huashanica*. From the selfing progenies of the monosomic addition line, we identified one individual plant, DA26, via GISH analysis containing 44 chromosomes, of which two chromosomes were from *P. huashanica* (Figure 1a,b). Three oligo probes, pSc200, pTa71A-2, and Oligo-44, were used to detect homoeologous groups of the alien chromosomes through FISH analysis (Figure 1c,d). The result showed that the exogenous chromosome was 4Ns of *P. huashanica* with red pSc200 signals at double-ended chromosomes and no pTa71A-2 and Oligo-44 hybridization patterns [27]. This result conferred that the additional line DA26 carried two 4Ns chromosomes.

In one selfing generation of DA26, from the 100 individuals (Figure 2), we identified only 29 plants containing two exogenous 4Ns chromosomes and 35 plants containing one exogenous chromosome. In contrast, other 36 plants did not contain any exogenous chromosomes. This result suggested that the DA26 was not stable cytologically and showed segregation in the following generations.

### 3.2. The 4Ns Additional Line Showed High Resistance to FHB

Based on the GISH analysis, we selected the 4Ns additional plants, transferred them to the greenhouse, and infected them with *Fg* strain PH-1. The results showed that the 4Ns additional line DA26 was highly resistant to *Fg* strain PH-1, with a similar phenotype as Sumai 3. However, the parental lines CS and CS*ph2b*, as well as the susceptible control “Roblin”, showed high susceptibility to *Fg* strain PH-1 (Figure 3a,d). We also checked the performance of these lines in the field with the artificial inoculation of *Fg* strain PH-1 and a mixed Sichuan *Fg* conidial suspension at the flowering stage (Figure 3b,c,e,f). The results were the same as in the greenhouse conditions (Figure 3a,d), supporting that the 4Ns additional line DA26 contained the FHB resistance gene(s) from the chromosome 4Ns of *P. huashanica*.

### 3.3. Detection of Exogenous Genes for FHB Resistance

We used link markers or functional markers of *Fhb3*, *Fhb6*, and *Fhb7* for gene identification in *P. huashanica* and DA26. The result showed that the positive controls *L. racemosus*, *C. kamoji*, and *Th. ponticum* amplified the target bands corresponding to *Fhb3*, *Fhb6* and *Fhb7*, while the negative controls (CS and CS*ph2b*) did not. As with the negative controls, DA26 did not contain these known *Fhb* genes, indicating that the 4Ns chromosomes from *P. huashanica* may carry a novel *Fhb* gene (Figure 4).

### 3.4. The Specific Markers of Chromosome 4Ns for Wheat Breeding Selection

Since no genome source was available for *P. huashanica*, we used the genome of *Leymus chinensis* (accession: Lc6-5, NsNsXmXm) to develop the specific markers for the chromosome 4Ns, since they contain similar Ns genome contents [33]. We compared the group 4 chromosomes of common wheat reference genomes with the 4Ns genome of *L. chinensis* and selected 1076 genes, which showed conceived localizations and sequence order. Based on the specific sequence of *Leymus chinensis* 4Ns chromosomes, we developed 103 primers, and 8 primers underwent specific amplification on the *P. huashanica* and 4Ns additional line, suggesting these markers could be used for 4Ns selection (Figure 5; Appendix A). In addition, seven markers could be amplified from the *P. huashanica* but not from the 4Ns additional line. These results suggested that *Leymus chinensis* (NsNsXmXm) could provide a reference Ns genome for *P. huashanica* but contains some variations. The eight markers developed in this study, six targeted at the 4NsS and two targeted at the 4NsL, could be used for gene mapping or marker-associated selection in future breeding systems.

### 3.5. The FHB Resistance and Marker Linkage Analysis in DA26 Selfing Line

Cytological identification was performed on 72 selfed progenies of DA26 using genomic in situ hybridization (GISH) technology (Figure 6a–d). Meanwhile, molecular marker analysis was carried out for all individual plants with the specific molecular markers *Lc032331-2* and *Lc034517-3* (Figure 6e). The results showed that specific band of the expected size could be amplified in all 49 DA26(+) plants carrying the exogenous 4Ns chromosome, while the target band did not appear in the 23 DA26(−) plants without the 4Ns chromosome. At the same time, under greenhouse conditions, phenotypic identification was conducted using the strain PH-1 (Figure 6f,g). All the 49 progenies of DA26(+) carrying exogenous 4Ns chromosomes exhibited stable high resistance, while these 23 DA26(−) plants showed high susceptibility. Therefore, it can be concluded that segregation between resistant and susceptible phenotypes occurred in the selfed progenies, which is consistent with the results of cytological identification.

### 3.6. Agronomic Performance of 4Ns Additional Line

To evaluate the agronomic traits of the 4Ns additional line, we planted the CS, CS*ph2b*, and the disomic addition progenies of the DA26 line in the field in normal conditions without disease infection (Figure 7). The screening of agronomic traits showed that, compared to the wheat parents, the plant height of DA26 was significantly reduced, while there were no significant differences in the other agronomic traits (Table 1). This result suggested that the 4Ns of *P. huashanica* has no obvious linkage drag in the common wheat background, showing great potential for wheat resistance breeding.

## 4. Discussion

As a tertiary gene pool for wheat, *P. huashanica* harbors abundant genes related to stress resistance and other good agronomic traits, establishing this species as a vital genetic resource for wheat improvement through wide hybridization with common wheat. For example, the 1Ns addition line shows resistance to wheat powdery mildew and leaf rust [34,35]; the 2Ns addition line is resistant to stripe rust and take-all disease [36,37]; the 3Ns, 4Ns, and 5Ns addition lines all display resistance to stripe rust [38,39,40]. The 6Ns addition line shows high resistance to stripe rust and features profuse flowering, high grain number per spike, and early maturity [41,42]. The 7Ns addition line combines resistance to powdery mildew and leaf rust with early maturity characteristics [43,44]. Recent research shows that the wheat–*P*. *huashanica* 1Ns long-arm ditelosomic addition lines, 2Ns(2D) substitution line, and 3Ns additional line exhibited enhanced resistance to FHB [23,45]. Unfortunately, only a few wheat–*P. huashanica*-derived lines with FHB resistance are available for wheat breeding. In this study, the disomic addition line DA26 of the wheat–*P. huashanica* 4Ns chromosome was successfully developed. Phenotypic analysis showed that DA26 exhibited significant type II resistance (resistance to spread) to FHB and had no obvious linkage drag (Figure 7). However, DA26 is genetically unstable, which may be attributed to the low transmission efficiency of exogenous chromosomes during gametogenesis. This phenomenon has been observed in some cases of exogenous chromosomes introduced into the wheat background, such as the E chromosomes of *Th. elomgatum* and the 3V chromosome of *D. villosum* [46,47]. Since DA26 is not genetically stable yet, our subsequent research plan is to develop small-segment translocation lines through radiation mutagenesis or hybridization with the CS*ph1b* mutant.

Wheat-related species exhibit significant potential for genetic improvements in FHB resistance in wheat [48]. However, progress in wheat FHB resistance breeding has been slow, primarily due to the scarcity of resistance sources. To date, two resistance genes, *Fhb1* and *Fhb7*, have been cloned. *Fhb1*, derived from Sumai 3 and Wangshuibai, is located on chromosome 3BS, the causal gene *TaHRC* (encoding a histidine-rich calcium-binding protein) has been widely utilized in global breeding programs [10]. *Fhb7*, originating from *Th. ponticum* chromosome 7E, was cloned by Wang et al. (2020) and encodes a glutathione S-transferase (GST), demonstrating significant breeding potential [8]. Meanwhile, by identifying alien chromosome translocation lines, the FHB resistance gene *Fhb6* was mapped to chromosome 1Ets#1S of *C. kamoji* [32]. Additionally, the FHB resistance genes *FhbRc1* and *FhbRc2* were localized to chromosomes 7ScL and 4ScL of *R. ciliaris*, respectively, further enriching the genetic resources for wheat FHB resistance breeding [18,49]. In this study, we developed an FHB-resistant addition line, and linkage analysis revealed that the 4Ns chromosome from *P*. *huashanica* confers FHB resistance and carries a novel resistance gene (Figure 4 and Figure 6). This provides a novel resistant germplasm for wheat resistance breeding and highlights the significant value of *P. huashanica* in wheat improvement.

Molecular markers play a crucial role in tracking alien chromatin that carries elite genes within a wheat background, thereby significantly improving the selection efficiency [50]. Despite identifying many valuable traits in *P. huashanica*, the lack of specific, stable, efficient, and reliable markers limits their breeding application. To improve the stability and specificity of the markers, subsequent studies developed Sequence-Characterized Amplified Region (SCAR) markers targeting the 1Ns, 2Ns, 3Ns, and 5Ns chromosomes based on RAPD and Expressed Sequence Tag–Sequence Tagged Site (EST–STS) markers [51,52,53]. We previously developed 163 specific molecular markers for the 7Ns chromosome using Genotyping-by-Sequencing (GBS) data and Specific Locus-Amplified Fragment Sequencing (SLAF-seq) [20,43]. However, there are very few reports on the molecular marker development of 4Ns chromosome of *P. huashanica*. In this study, we utilized the publicly available whole-genome sequence of *Leymus chinensis* (Lc6-5) [33] and successfully developed eight specific molecular markers for tracking the 4Ns chromosome (Figure 5). The eight markers developed in this study can quickly track the 4Ns chromosome of *P. huashanica* in the common wheat genetic background, which can facilitate future wheat resistance breeding progress and FHB resistance gene mapping.

## Figures and Tables

**Figure 1 plants-14-01104-f001:**
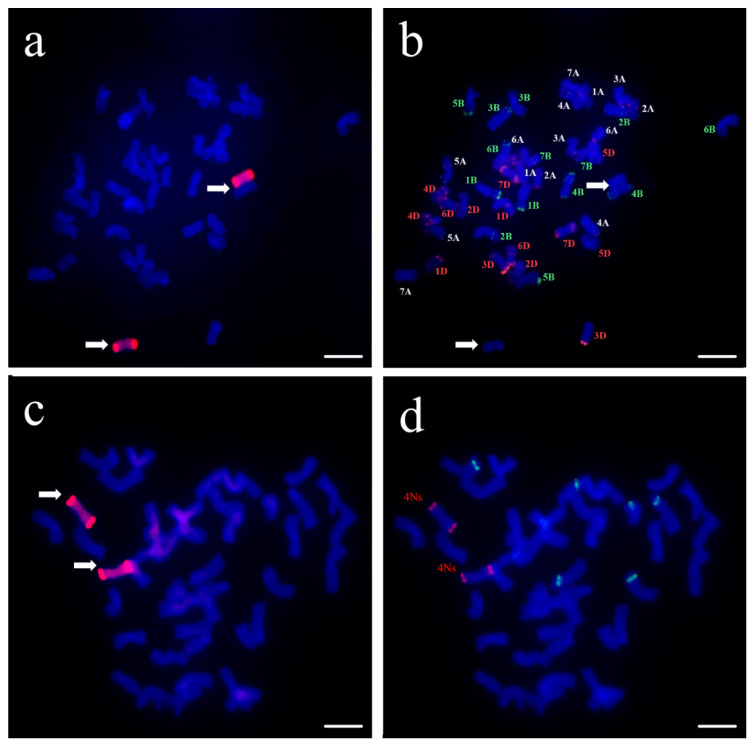
GISH and FISH analysis of the wheat–*Psathyrostachys huashanica* disomic addition line DA26. (**a**): GISH analysis of DA26, *P. huashanica* genomic DNA as a probe (red); (**b**): FISH analysis of DA26, Oligo-pSc119.2 (green) and Oligo-pTa535 (red) as probes, (**a**,**b**) are the same metaphase cells of the disomic addition line DA26. (**c**): GISH analysis of DA26, *P. huashanica* genomic DNA as a probe (red); (**d**): FISH analysis of DA26, Oligo-pSc200 (red), Oligo-pTa71A-2 (green), Oligo-44 (yellow), (**c**,**d**) are the same metaphase cells of the disomic addition line DA26. Arrows indicate the introduced *P. huashanica* chromosomes in line DA26. Scale bars = 10 μm.

**Figure 2 plants-14-01104-f002:**
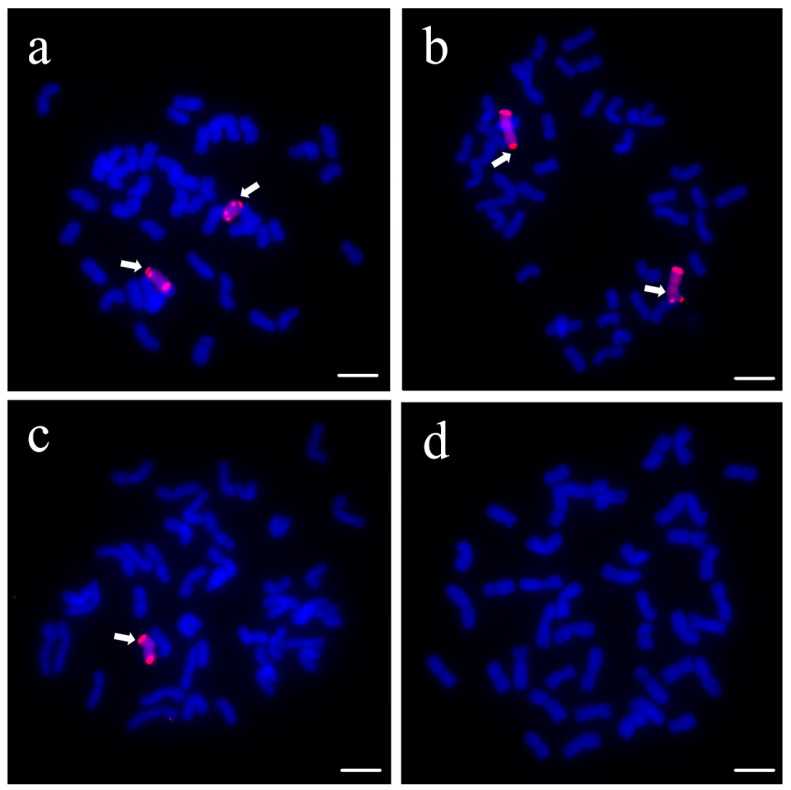
GISH analysis of selfed progenies of DA26 of Wheat-*Psathyrostachys huashanica* disomic addition line. (**a**,**b**) contain two exogenous 4Ns chromosomes; (**c**) contains one exogenous 4Ns chromosome; (**d**) contains no exogenous chromosome. Arrows indicate the introduced *P. huashanica* chromosomes in line DA26. Scale bars = 10 μm.

**Figure 3 plants-14-01104-f003:**
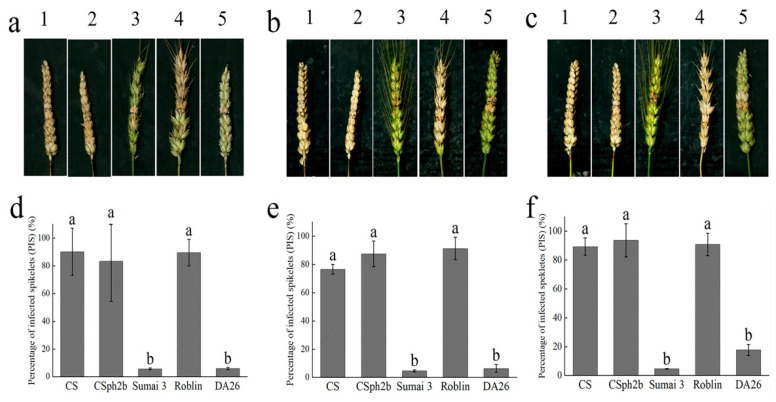
FHB resistance performance of 4Ns additional line and the parental lines. (**a**,**d**): Identification of resistance to FHB using strain PH-1 in greenhouse condition. (**b**,**e**): Identification of resistance to FHB using strain PH-1 in field condition. (**c**,**f**): Identification of resistance to FHB using mixed Sichuan local physiological races in field condition. In (**a**–**c**), from left to right, are 1: CS, 2: CSp*h2b*, 3: Sumai 3, 4: Roblin, 5: DA26. (**d**–**f**) Different letters denote significant differences (*p* ≤ 0.05) of the mean values of the percentage of infected spikes (PISs) in different wheat accessions. *n* = 5 for each line.

**Figure 4 plants-14-01104-f004:**
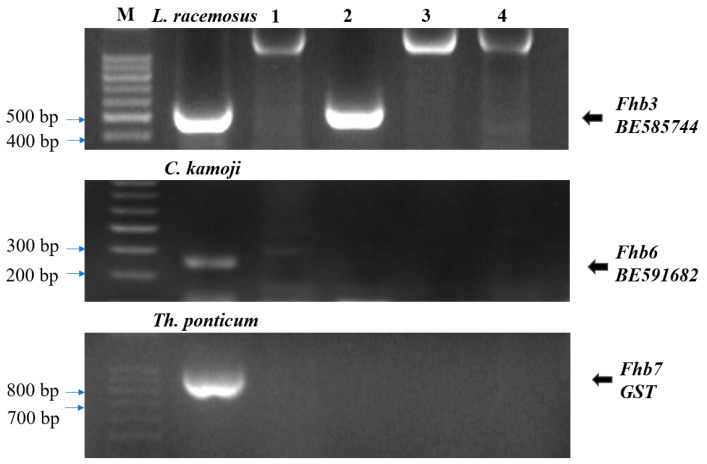
Molecular marker detection of known *Fhb* genes in wheat-related species. M: DNA marker (1500 bp). 1: CS; 2: *P. huashanica*; 3: CS*ph2b*; 4: DA26.

**Figure 5 plants-14-01104-f005:**
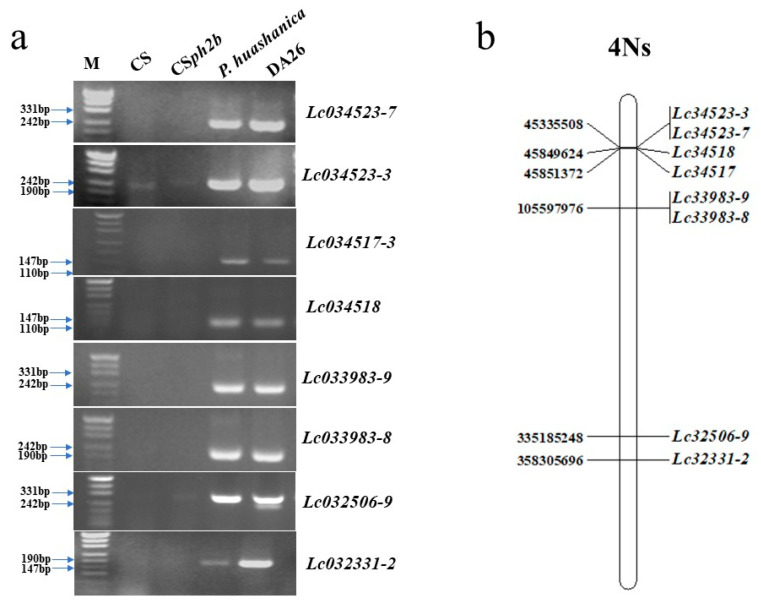
Development of 4Ns-specific molecular markers. (**a**): The PCR amplification patterns of 4Ns-specific molecular markers. M: DNA marker (500 bp). (**b**): Physical mapping of 4Ns-specific molecular markers. The sequence of chromosome 4Ns of *Leymus chinese* was used as reference. The genes corresponding to molecular markers were present in the right side of the 4Ns chromosome, and the gene positions were present in the left side of the 4Ns chromosome.

**Figure 6 plants-14-01104-f006:**
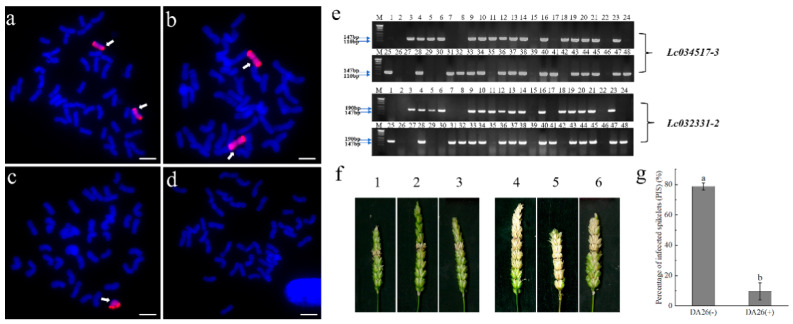
The FHB resistance and marker linkage analysis in DA26 selfing line. (**a**–**d**): GISH identification of 72 selfed progenies of DA26. (**e**): Molecular marker verification of DA26 selfed progenies, M: DNA marker (500 bp); 1: CS, 2: CS*ph2b*, 3: *Psathyrostachys huashanica*, 4-48: selfed progenies of DA26. (**f**): Identification results of FHB resistance of selfed progenies of DA26 in the greenhouse condition, including 1-3 DA26 (+), 4-6 DA26 (−), the designation “+” indicates the positive plants carrying *P. huashanica* 4Ns chromosome, while “−” indicates the negative plants lacking *P. huashanica* 4Ns chromosome. (**g**): Statistics of percentage of infected spikes (PISs) of DA26 selfed progenies. Arrows indicate the introduced *P. huashanica* chromosomes in DA26 selfing lines (**a**–**c**). Different letters denote significant differences (*p* ≤ 0.05) of the mean values of the percentage of infected spikes (PISs) in DA26 (+) and DA26 (−) groups (**g**).

**Figure 7 plants-14-01104-f007:**
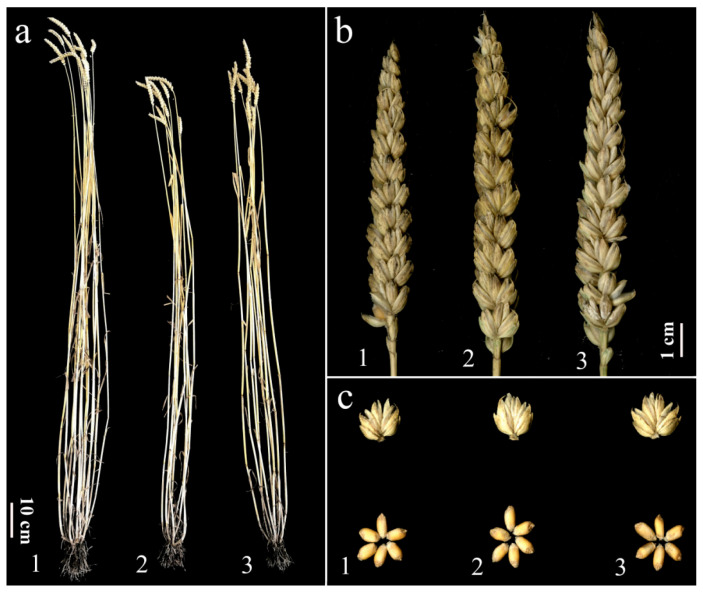
The agronomic traits of the wheat–*Psathyrostachys huashanica* disomic addition line DA26 and its wheat parents. (**a**): plant height, (**b**): spike length, (**c**): spikelet, grain; 1: CS, 2: CS*ph2b*, 3: DA26.

**Table 1 plants-14-01104-t001:** Agronomic traits of DA26, and wheat parent CS, CS*ph2b*.

Materials	Plant Height (cm)	Tiller	Spike Length (cm)	The Number of Spikelets	Grains per Spike	1000 Grain Weight (g)
CS	144.17 ± 1.26 a	10.79 ±1.17 a	9.86 ± 0.29 a	24.79 ± 0.44 a	57.21 ± 1.20 a	23.05 ± 0.05 a
CS*ph2b*	141.76 ± 0.97 ab	10.21 ±0.33 a	9.56 ± 0.18 a	24.07 ± 0.49 a	54.43 ± 1.90 a	22.69 ± 0.13 a
DA26	138.30 ± 2.33 b	11.57 ±1.29 a	9.34 ± 0.27 a	23.07 ± 0.68 a	55.07 ± 2.45 a	22.50 ± 0.71 a

Note: Data in the columns indicates means ± standard errors. Lowercase letters following the means indicate significant differences at the *p* ≤ 0.05 levels.

## Data Availability

The datasets generated and/or analyzed during the current study are available from the corresponding author upon reasonable request.

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
