# Peer review of "Wheat-Psathyrostachys huashanica 4Ns Additional Line Confers Resistance to Fusarium Head Blight"

_plants, 2025, doi:10.3390/plants14071104_

Round 1
Reviewer 1 Report
Comments and Suggestions for Authors
The manuscript presents an investigation of the 4Ns additional wheat line (DA26) derived from Psathyrostachys huashanica and its resistance to Fusarium Head Blight (FHB). The authors used cytogenetic (GISH/FISH) analysis to confirm the presence of 4Ns chromosomes and qPCR-based markers to track its genetic inheritance. Field and greenhouse trials demonstrated strong Type II FHB resistance. Agronomic evaluations showed DA26 had a shorter plant height but no other significant trade-offs in yield-related traits. The study introduces new molecular markers to assist in wheat breeding programs.
1. General Evaluation
This manuscript addresses a highly relevant topic in wheat breeding: resistance to Fusarium Head Blight (FHB), a disease that significantly compromises global wheat production and food security. By focusing on the introgression of genetic material from Psathyrostachys huashanica, a wild wheat relative, into a wheat background (DA26 line), the study introduces important insights into the search for novel sources of FHB resistance.
One of the most important strengths of this study is its alignment with current priorities in sustainable agriculture and breeding for disease resistance. The manuscript's scientific merit lies in its effort to characterize DA26, a wheat line incorporating P. huashanica chromatin, and to demonstrate its high level of FHB resistance, both under controlled (greenhouse) and natural (field) conditions. The Authors employ robust cytogenetic analyses, including GISH and FISH, to confirm chromosomal introgression, and they support their phenotypic assessments with qPCR-based validation of Fusarium presence, strengthening the scientific bases of their investigations. Furthermore, the development of molecular markers linked to the alien chromatin enhances the potential for marker-assisted selection (MAS), offering practical tools for future breeding programs.
However, despite these strengths, several critical concerns need to be addressed to fully validate the study's findings and enhance its suitability for publication in a high-impact journal such as Plant.
2. Major Concerns and Recommended Revisions
Although the study presents a solid initial characterization of DA26, a number of relevant aspects remain underexplored or insufficiently addressed, which may limit the broader impact and application of the results.
One major concern is the lack of genetic characterization of the resistance mechanism underlying the FHB resistance observed in DA26. While the phenotypic resistance is clearly documented, the manuscript does not investigate the genetic basis of this resistance nor attempt to identify candidate genes or quantitative trait loci (QTLs). This omission limits the mechanistic understanding of DA26's resistance and its potential use in marker-assisted breeding. Given that high-impact journals increasingly value studies that combine phenotypic assessments with insights into genetic and molecular mechanisms, the absence of such analysis is a critical gap. A discussion of putative genes or known resistance QTLs potentially associated with the introgressed segment would enhance the manuscript's scientific depth. Additionally, future perspectives should consider RNA-seq or transcriptomic analyses aimed at identifying defense-related genes and pathways, which would provide a comprehensive understanding of DA26's resistance mechanism and its potential stability across environments.
Another concern is the limited and poorly defined sample size used in field trials and qPCR validation. The authors do not clearly state how many biological replicates were used in disease resistance scoring or molecular assays. This lack of detail raises concerns regarding the statistical robustness and reproducibility of the findings. Field resistance, particularly for complex diseases like FHB, is influenced by environmental variability, and limited replication may not capture this complexity. The authors should explicitly describe the number of plants analyzed per line and per treatment, and if the sample size is indeed small, this limitation should be acknowledged and addressed with proposals for future larger-scale validations.
A further issue that undermines the practical value of DA26 is the segregation and stability of the introgression. The manuscript mentions that DA26 segregates in selfed progenies, suggesting cytogenetic instability. This aspect is critical because any line intended for breeding must exhibit chromosomal stability to ensure consistent inheritance of resistance traits. The authors should elaborate on possible strategies to stabilize DA26, such as radiation mutagenesis, backcrossing, or selection of stable derivatives. Furthermore, they should discuss whether this observed instability impacts the resistance phenotype across generations and how this challenge could be overcome in breeding programs.
Another important limitation is the lack of agronomic performance data, especially concerning grain yield and quality. While DA26 is proposed as a candidate for breeding, no information is provided on whether the line maintains acceptable yield, grain morphology, or other essential traits in addition to its disease resistance. This is a fundamental concern for breeders, as alien introgressions often bring yield penalties or undesirable traits. Without yield data, the claim that DA26 is ready for breeding adoption appears overstated. Including at least preliminary data on yield, grain quality, or other agronomic traits would significantly strengthen the manuscript. If such data are not yet available, this limitation should be transparently acknowledged, and the need for further multi-environmental trials should be outlined.
3. Minor concerns and suggested revisions
In addition to these major issues, there are several minor concerns regarding contextualization within existing literature, and regulatory considerations, which, if addressed, would greatly improve the manuscript's readability and relevance.
Moreover, the discussion lacks adequate comparison with existing FHB resistance sources, such as Fhb1, Fhb7, and other well-characterized genes or QTLs used in breeding. Including a comparative analysis, possibly in tabular form, would situate DA26's resistance relative to other known sources and provide breeders with a clearer understanding of its potential advantages or limitations. Such a discussion would also underscore the novelty and added value of using P. huashanica chromatin compared to traditional sources of resistance.
Additionally, the regulatory aspects of deploying alien chromatin in breeding programs are not discussed. Given that the use of wild species in breeding may raise biosafety and regulatory concerns, a brief acknowledgment of these issues would strengthen the manuscript's practical relevance. Addressing whether the alien chromosomal material in DA26 could face regulatory hurdles would provide a more realistic perspective for breeders interested in its adoption.
4. Conclusion
In summary, while the manuscript presents a valuable and timely contribution to the field of wheat disease resistance, particularly regarding the novel use of Psathyrostachys huashanica for FHB resistance, several critical revisions are necessary to enhance its scientific rigor and practical applicability. Addressing the genetic mechanisms of resistance, clarifying sample sizes, ensuring chromosomal stability, and presenting agronomic data are essential steps to fully validate the utility of DA26 as a breeding resource. Improvements in language clarity, comparison with existing resistance sources, and attention to regulatory considerations will further increase the manuscript’s quality and impact. With these revisions, the study will offer significant advances in breeding for FHB resistance and contribute meaningfully to sustainable wheat production.

From a linguistic and organizational perspective, certain sentences are overly long and convoluted, which can hinder comprehension. For example, the sentence describing the resistance comparison between DA26 and its parental lines could be reformulated for clarity. A clearer phrasing, such as "DA26 exhibited strong resistance to FHB under both greenhouse and field conditions, whereas its parental lines, Chinese Spring and CSph2b, were highly susceptible," would improve readability. A general language revision aimed at simplifying complex sentences and improving flow is recommended to enhance accessibility for a broader scientific audience.
Author Response
The manuscript presents an investigation of the 4Ns additional wheat line (DA26) derived from Psathyrostachys huashanica and its resistance to Fusarium Head Blight (FHB). The authors used cytogenetic (GISH/FISH) analysis to confirm the presence of 4Ns chromosomes and qPCR-based markers to track its genetic inheritance. Field and greenhouse trials demonstrated strong Type II FHB resistance. Agronomic evaluations showed DA26 had a shorter plant height but no other significant trade-offs in yield-related traits. The study introduces new molecular markers to assist in wheat breeding programs.
General Evaluation
This manuscript addresses a highly relevant topic in wheat breeding: resistance to Fusarium Head Blight (FHB), a disease that significantly compromises global wheat production and food security. By focusing on the introgression of genetic material from Psathyrostachys huashanica, a wild wheat relative, into a wheat background (DA26 line), the study introduces important insights into the search for novel sources of FHB resistance. One of the most important strengths of this study is its alignment with current priorities in sustainable agriculture and breeding for disease resistance. The manuscript's scientific merit lies in its effort to characterize DA26, a wheat line incorporating P. huashanica chromatin, and to demonstrate its high level of FHB resistance, both under controlled (greenhouse) and natural (field) conditions. The Authors employ robust cytogenetic analyses, including GISH and FISH, to confirm chromosomal introgression, and they support their phenotypic assessments with qPCR-based validation of Fusarium presence, strengthening the scientific bases of their investigations. Furthermore, the development of molecular markers linked to the alien chromatin enhances the potential for marker-assisted selection (MAS), offering practical tools for future breeding programs. However, despite these strengths, several critical concerns need to be addressed to fully validate the study's findings and enhance its suitability for publication in a high-impact journal such as Plant.
Thanks a lot for your valuable comments, we improved our manuscript based on reviewers’ comments.
Major Concerns and Recommended Revisions
- Although the study presents a solid initial characterization of DA26, a number of relevant aspects remain underexplored or insufficiently addressed, which may limit the broader impact and application of the results. One major concern is the lack of genetic characterization of the resistance mechanism underlying the FHB resistance observed in DA26. While the phenotypic resistance is clearly documented, the manuscript does not investigate the genetic basis of this resistance nor attempt to identify candidate genes or quantitative trait loci (QTLs). This omission limits the mechanistic understanding of DA26's resistance and its potential use in marker-assisted breeding. Given that high-impact journals increasingly value studies that combine phenotypic assessments with insights into genetic and molecular mechanisms, the absence of such analysis is a critical gap. A discussion of putative genes or known resistance QTLs potentially associated with the introgressed segment would enhance the manuscript's scientific depth. Additionally, future perspectives should consider RNA-seq or transcriptomic analyses aimed at identifying defense-related genes and pathways, which would provide a comprehensive understanding of DA26's resistance mechanism and its potential stability across environments.
Response: Linkage analysis on a genetic segregation population derived from self-pollination of DA26, revealed the FHB resistance co-segregated with the alien 4Ns chromosome. To date, no studies have reported the mapping or cloning of FHB resistance genes on the 4Ns chromosome. In addition, we added a experiment to exclude the possibility that 4Ns carries the known FHB resistance genes (Fhb3, Fhb6, and Fhb7) by using diagnostic molecular markers (Fig. 4). This confirms that 4Ns carries a novel FHB resistance gene. However, due to the difficulty in recombination between the 4Ns chromosome and common wheat chromosomes, fine mapping of the resistance gene or QTLs is challenging. We plan to address this by using two approaches: radiation treatment of DA26 and introducing the ph1b mutation to promote recombination. We will also use RNA-seq or transcriptomic analyses to identifying defense-related genes and pathways in our future studies.
- Another concern is the limited and poorly defined sample size used in field trials and qPCR validation. The authors do not clearly state how many biological replicates were used in disease resistance scoring or molecular assays. This lack of detail raises concerns regarding the statistical robustness and reproducibility of the findings. Field resistance, particularly for complex diseases like FHB, is influenced by environmental variability, and limited replication may not capture this complexity. The authors should explicitly describe the number of plants analyzed per line and per treatment, and if the sample size is indeed small, this limitation should be acknowledged and addressed with proposals for future larger-scale validations.
Response: We conducted a total of three resistance evaluations for DA26 and its parental lines: the first was a FHB resistance assessment using the PH-1 strain in the greenhouse, the second was an FHB resistance assessment using the PH-1 strain in the field, and the third was an FHB resistance assessment using a mixed physiological race from Sichuan Province in the field. Five plants per line were evaluated, and the results are detailed in 3.2. For the 72 selfed progenies of DA26, 49 plants contained the 4Ns and all showed high resistance to FHB, 23 plants did not contained the 4Ns and all showed high susceptible to FHB. This linkage analysis also support a kind of replicates of 4Ns conferring of FHB resistance. We added these detailed information in 3.5.
- A further issue that undermines the practical value of DA26 is the segregation and stability of the introgression. The manuscript mentions that DA26 segregates in selfed progenies, suggesting cytogenetic instability. This aspect is critical because any line intended for breeding must exhibit chromosomal stability to ensure consistent inheritance of resistance traits. The authors should elaborate on possible strategies to stabilize DA26, such as radiation mutagenesis, backcrossing, or selection of stable derivatives. Furthermore, they should discuss whether this observed instability impacts the resistance phenotype across generations and how this challenge could be overcome in breeding programs.
Response: Chromosomal stability is a critical factor for the practical value of breeding materials. In response, we would like to clarify that DA26, as an intermediate material, will be stabilized through strategies such as radiation mutagenesis, suppression of the Ph1 gene, or backcross breeding to promote the stable integration of alien chromosomes into the wheat genome. In the future, we will focus on developing small-segment translocation lines to achieve stable inheritance of resistance genes and their application in breeding.
- Another important limitation is the lack of agronomic performance data, especially concerning grain yield and quality. While DA26 is proposed as a candidate for breeding, no information is provided on whether the line maintains acceptable yield, grain morphology, or other essential traits in addition to its disease resistance. This is a fundamental concern for breeders, as alien introgressions often bring yield penalties or undesirable traits. Without yield data, the claim that DA26 is ready for breeding adoption appears overstated. Including at least preliminary data on yield, grain quality, or other agronomic traits would significantly strengthen the manuscript. If such data are not yet available, this limitation should be transparently acknowledged, and the need for further multi-environmental trials should be outlined.
Response: Currently, DA26 is an addition line, and the genetic background is Chinese Spring (CS), which has bad agronomic performance. However, compare to CS, the DA26 does not show obvious linkage drag (Fig. 7 and Table 1). In this stage, we focus on the FHB resistance of DA26. In the future, we plan to create translocation lines in better wheat varieties genetic background, especially small-fragment translocation lines, for improving the grain yield and quality.
Minor concerns and suggested revisions
In addition to these major issues, there are several minor concerns regarding contextualization within existing literature, and regulatory considerations, which, if addressed, would greatly improve the manuscript's readability and relevance. Moreover, the discussion lacks adequate comparison with existing FHB resistance sources, such as Fhb1, Fhb7, and other well-characterized genes or QTLs used in breeding. Including a comparative analysis, possibly in tabular form, would situate DA26's resistance relative to other known sources and provide breeders with a clearer understanding of its potential advantages or limitations. Such a discussion would also underscore the novelty and added value of using P. huashanica chromatin compared to traditional sources of resistance.
Response: We provided a detailed description of the current research progress on wheat Fusarium head blight (FHB) resistance genes, including the origins, localization, and breeding applications of Fhb1, Fhb7, Fhb6, FhbRc1, and FhbRc2 in discussion part. We added additional experiment, and excluded the possibility that 4Ns carries the known FHB resistance genes (Fhb3, Fhb6, and Fhb7) by using diagnostic molecular markers (Fig.4).
Additionally, the regulatory aspects of deploying alien chromatin in breeding programs are not discussed. Given that the use of wild species in breeding may raise biosafety and regulatory
concerns, a brief acknowledgment of these issues would strengthen the manuscript's practical relevance. Addressing whether the alien chromosomal material in DA26 could face regulatory hurdles would provide a more realistic perspective for breeders interested in its adoption.
Response: In the current field of wheat breeding, there are already mature examples of using the alien chromosomes of Psathyrostachys huashanica for variety cultivation. For instance, Xinong 501 was approved at the national level in 2020, and Xinong 68 was approved at the provincial level in Shaanxi in 2023. These varieties, which all utilize the alien chromosomes of Psathyrostachys huashanica, have been widely recognized and popularized. We added these example in discussion part. This indicates that the existing regulatory system can effectively evaluate and accept this type of breeding method. The pattern of DA26's utilization of the alien chromosomes of Psathyrostachys huashanica is similar to that of the above-mentioned successful varieties, so the possibility of it facing regulatory obstacles is low.
Conclusion
In summary, while the manuscript presents a valuable and timely contribution to the field of wheat disease resistance, particularly regarding the novel use of Psathyrostachys huashanica for FHB resistance, several critical revisions are necessary to enhance its scientific rigor and practical applicability. Addressing the genetic mechanisms of resistance, clarifying sample sizes, ensuring chromosomal stability, and presenting agronomic data are essential steps to fully validate the utility of DA26 as a breeding resource. Improvements in language clarity, comparison with existing resistance sources, and attention to regulatory considerations will further increase the manuscript’s quality and impact. With these revisions, the study will offer significant advances in breeding for FHB resistance and contribute meaningfully to sustainable wheat production.
Thanks a lot for your valuable comments.

Reviewer 2 Report
Comments and Suggestions for Authors
In the study entitled "Wheat-Psathyrostachys huashanica 4Ns additional line confers resistance to Fusarium head blight," Li et al. investigated the potential of a wheat-Psathyrostachys huashanica derived addition line as a genetic resource for improving resistance to Fusarium head blight (FHB) in wheat. They identified the P. huashanica chromosome 4Ns disomic additional line, DA26, using genomic in situ hybridization (GISH) and fluorescence in situ hybridization (FISH) analyses. DA26 exhibited high resistance, similar to Sumai-3, to FHB in both greenhouse and field conditions, while the checks and susceptible parents showed high susceptibility. Additionally, DA26 had no significant differences in agronomic traits other than plant height, which also showed no linkage drag. The authors also developed eight P. huashanica – specific molecular markers to facilitate easy detection of chromosome 4Ns.
To me, this is a very important study and has some merits particularly for future FHB resistance breeding since there are not too many major genes identified for FHB resistance in wheat. Strengths of the study include:
- Identification of the 4Ns additional line DA26 with high resistance to FHB.
- Use of GISH and FISH analyses for detection and identification of additional chromosomes.
- Development of P. huashanica– specific primers for detection of chromosome 4Ns in wheat background.
Despite its strengths, I also have some major concerns about this study. Particularly, about the use and integration of the chromosome 4Ns of P. huashanica into commercially viable wheat cultivars. The identified disomic addition line DA26 is identified from the previously developed wheat (CSph2b) × P. huashanica hybrids (Kang et al. 2008) and is not a genetically stable line. Furthermore, Kang et al. (2008) previously detected the non-homoeology between the genomes of Triticum aestivum L. (ABD) and P. huashanica (Ns), presenting a significant hurdle for breeding programs. Although the authors briefly mentioned their subsequent research plan to develop translocation lines by radiation mutagenesis or probably by crossing with CSph1b mutants, both approaches pose considerable challenges. Have the authors considered using null alleles of the Ph1 gene or employing PhI, an epistatic inhibitor of the Ph1 gene from Aegilops speltoides Tausch, to facilitate homeologous chromosome pairing and recombination to achieve genetically compensating transfers?
I also have a few minor concerns. The methods used to develop the DA26 line are not thoroughly described. Despite citing your previous study (Kang et al. 2008), a more detailed methodology would enhance reproducibility and clarity. While the developed primers are valuable for detecting chromosome 4Ns, they are not specific to FHB resistance. The authors should avoid suggesting their use in FHB resistance breeding as they do not confer resistance directly. Please refer to the attached PDF for further minor comments and suggestions.
Overall, this study provides valuable insights into the potential of wild relatives of wheat for FHB resistance. However, addressing the concerns outlined above would strengthen the manuscript and improve its applicability to practical breeding programs.

Author Response
Review 2:
In the study entitled "Wheat-Psathyrostachys huashanica 4Ns additional line confers resistance to Fusarium head blight," Li et al. investigated the potential of a wheat-Psathyrostachys huashanica derived addition line as a genetic resource for improving resistance to Fusarium head blight (FHB) in wheat. They identified the P. huashanica chromosome 4Ns disomic additional line, DA26, using genomic in situ hybridization (GISH) and fluorescence in situ hybridization (FISH) analyses. DA26 exhibited high resistance, similar to Sumai-3, to FHB in both greenhouse and field conditions, while the checks and susceptible parents showed high susceptibility. Additionally, DA26 had no significant differences in agronomic traits other than plant height, which also showed no linkage drag. The authors also developed eight P. huashanica – specific molecular markers to facilitate easy detection of chromosome 4Ns.
To me, this is a very important study and has some merits particularly for future FHB resistance breeding since there are not too many major genes identified for FHB resistance in wheat. Strengths of the study include:
- Identification of the 4Ns additional line DA26 with high resistance to FHB.
- Use of GISH and FISH analyses for detection and identification of additional chromosomes.
- Development of P. huashanica– specific primers for detection of chromosome 4Ns in wheat background.
Thank you very much for your valuable suggestions.
Despite its strengths, I also have some major concerns about this study. Particularly, about the use and integration of the chromosome 4Ns of P. huashanica into commercially viable wheat cultivars. The identified disomic addition line DA26 is identified from the previously developed wheat (CSph2b) × P. huashanica hybrids (Kang et al. 2008) and is not a genetically stable line. Furthermore, Kang et al. (2008) previously detected the non-homoeology between the genomes of Triticum aestivum L. (ABD) and P. huashanica (Ns), presenting a significant hurdle for breeding programs. Although the authors briefly mentioned their subsequent research plan to develop translocation lines by radiation mutagenesis or probably by crossing with CSph1b mutants, both approaches pose considerable challenges. Have the authors considered using null alleles of the Ph1 gene or employing PhI, an epistatic inhibitor of the Ph1 gene from Aegilops speltoides Tausch, to facilitate homeologous chromosome pairing and recombination to achieve genetically compensating transfers?
Response: The CSph1b mutants have been successfully used in our lab to develop the 7Ns translocation lines. The radiation mutagenesis is also a high efficiency method to create the translocation lines. We prefer to try these two methods firstly, if not succeed, we will try other methods in the future study.
I also have a few minor concerns. The methods used to develop the DA26 line are not thoroughly described. Despite citing your previous study (Kang et al. 2008), a more detailed methodology would enhance reproducibility and clarity. While the developed primers are valuable for detecting chromosome 4Ns, they are not specific to FHB resistance. The authors should avoid suggesting their use in FHB resistance breeding as they do not confer resistance directly. Please refer to the attached PDF for further minor comments and suggestions.
Overall, this study provides valuable insights into the potential of wild relatives of wheat for FHB resistance. However, addressing the concerns outlined above would strengthen the manuscript and improve its applicability to practical breeding programs.
Thank you very much for your valuable comments.

Reviewer 3 Report
Comments and Suggestions for Authors
I checked your manuscript and described comments below.
Fusarium head blight is a globally important disease affecting wheat and barley.
This paper provides an excellent study on the resistance of tetraploid Psathyrostachys huashanica Keng ex P. C. Kuo (2n = 2x = 14, NsNs) to Fusarium head blight.
I think you should consider the following points.
- If possible, it would be better to provide a table of the DNA sequences of the GFISH and Fish probes.
- The Ns reference genome data is available on figshare.com, but the URL/links doesn't open. I think you should check the URL/Links.
- I don't know the amount of DNA and the equipment used for PCR. I think it would be better to state that.
I don't think this paper has major problems and grammatical problems.
Author Response
Reviewer 3
I checked your manuscript and described comments below.
Fusarium head blight is a globally important disease affecting wheat and barley.
This paper provides an excellent study on the resistance of tetraploid Psathyrostachys huashanica Keng ex P. C. Kuo (2n = 2x = 14, NsNs) to Fusarium head blight.
I think you should consider the following points.
- If possible, it would be better to provide a table of the DNA sequences of the GFISH and Fish probes.
Response: The sequence information for the fluorescence in situ hybridization (FISH) probes is provided in Table S2.
- The Ns reference genome data is available on figshare.com, but the URL/links doesn't open. I think you should check the URL/Links.
Response: We corrected it in the text. The reference genome data of P. huashanica is based on the published whole - genome sequence of L. chinensis (Lc6 - 5). We have supplemented the relevant literature citation in the Materials and Methods section (2.6) to ensure the clear traceability of the data source.
Li, T.; Tang, S.; Li, W.; Zhang, S.; Wang, J.; Pan, D.; Lin, Z.; Ma, X.; Chang, Y.; Liu, B.; et al. Genome Evolution and Initial Breeding of the triticeae grass Leymus Chinensis dominating the eurasian steppe. Proc Natl Acad Sci 2023, 120, e2308984120, doi:10.1073/PNAS.2308984120.
- I don't know the amount of DNA and the equipment used for PCR. I think it would be better to state that.
Response: We supplemented the amount of DNA used in the PCR reaction (1 μL, 200 ng/μL) and the system and procedures employed in the Materials and Methods section (2.4and 2.6).
I don't think this paper has major problems and grammatical problems.
Thank you very much for your valuable comments.

Reviewer 4 Report
Comments and Suggestions for Authors
The manuscript reports the identification of a wheat-P. huashanica 4Ns disomic addition line and its characterization by the genomic in situ hybridization (GISH) and fluorescence in situ hybridization (FISH) techniques. Eight PCR markers specific for the P. huashanica 4Ns chromosome were developed. The disomic line expressed resistance to Fusarium head blight (FHB) in the greenhouse and field conditions, and showed a significant decrease of plant height and no significant differences in spike length, number of spikelets, grains per spike, 1000 grain weight compared to the control cultivar.
The title adequately describes the subject of the manuscript, and the abstract briefly tells what was done and summarizes the main results and conclusions. The author’s contribution is placed in its proper perspective concerning the state of knowledge. The subject is developed logically and effectively, and the manuscript is well-organized and concise. The conclusions are adequate and supported by the data.
The subject falls within the scope of “Plants”.
Minor revisions.
- The cytological analysis and the segregation between resistant and susceptible phenotypes in the selfed progenies indicated the disomic addition line was not stable.
Possible causes of this cytological and phenotypic instability should be reported in the Results section or the Conclusion.
- Figure 4 reports the genes corresponding to the developed molecular markers. Could these genes be considered candidate genes for resistance to Fusarium head blight? If so, report the function of these genes.
Author Response
Review 4
The manuscript reports the identification of a wheat-P. huashanica 4Ns disomic addition line and its characterization by the genomic in situ hybridization (GISH) and fluorescence in situ hybridization (FISH) techniques. Eight PCR markers specific for the P. huashanica 4Ns chromosome were developed. The disomic line expressed resistance to Fusarium head blight (FHB) in the greenhouse and field conditions, and showed a significant decrease of plant height and no significant differences in spike length, number of spikelets, grains per spike, 1000 grain weight compared to the control cultivar.
The title adequately describes the subject of the manuscript, and the abstract briefly tells what was done and summarizes the main results and conclusions. The author’s contribution is placed in its proper perspective concerning the state of knowledge. The subject is developed logically and effectively, and the manuscript is well-organized and concise. The conclusions are adequate and supported by the data.
The subject falls within the scope of “Plants”.
Thank you very much for your valuable comments.
Minor revisions.
- The cytological analysis and the segregation between resistant and susceptible phenotypes in the selfed progenies indicated the disomic addition line was not stable.
Possible causes of this cytological and phenotypic instability should be reported in the Results section or the Conclusion.
Response: Thank you for your valuable feedback on the stability issues of the DA26 additional line. We have supplemented the discussion section with possible reasons for the instability of DA26, primarily attributed to the low transmission efficiency of exogenous chromosomes during gamete formation. This phenomenon has been observed in various exogenous chromosomes introduced into the wheat background, such as the E chromosome of Thinopyrum elongatum and the 3V chromosome of Dasypyrum villosum. We have also included relevant references in the discussion section.
We plan to employ strategies such as radiation mutagenesis, suppression of the Ph1 gene, or backcross breeding to promote the stable integration of exogenous chromosomes with wheat chromosomes, thereby achieving stable inheritance of resistance genes and their application in breeding.
Once again, thank you for your suggestions. These improvements will significantly enhance the scientific rigor and practical value of the research.
- Zhang, J.; Jiang, Y.; Guo, Y.; Wang, Y.; Yang, Y.; Li, X.; et al. Transmission of Dasypyrum villosum3V Chromosome in Different Wheat Genetic Backgrounds. Plant Genet. Resour. 2021, 22(5), 1355–1364, doi.:10.13430/j.cnki.jpgr.20210427003.
- Li, H.; Liu, H.; Dai, Y.; Huang, S.; Zhang, J.; Gao, Y.; Chen, J. Transmission Characteristics of Thinopyrum elongatumE Chromosomes in Tetraploid Wheat Background. Yi Chuan 2016, 38(11), 1019–1028, doi:10.16288/j.yczz.16-107.
- Figure 4 reports the genes corresponding to the developed molecular markers. Could these genes be considered candidate genes for resistance to Fusarium head blight? If so, report the function of these genes.
Response: Thank you for your valuable comments. We performed collinearity analysis between genes from the Ns chromosome group of L. chinensis and the ABD genome of wheat. By randomly selecting genes with strong collinearity and conservation at different chromosomal locations, we inferred that these genes are likely to be present at the same positions in P. huashanica. Based on these criteria, we randomly selected genes and designed primers using the specific sequences of Leymus chinensis by aligning them with homologous genes on the wheat ABD chromosomes. Therefore, these primers were not developed based on potential FHB resistance candidate genes on the 4Ns chromosome.
